# Molecular Circuits of Immune Sensing and Response to Oncolytic Virotherapy

**DOI:** 10.3390/ijms25094691

**Published:** 2024-04-25

**Authors:** Darshak K. Bhatt, Toos Daemen

**Affiliations:** Department of Medical Microbiology and Infection Prevention, University Medical Center Groningen, University of Groningen, P.O. Box 30 001, HPC EB88, 9700 RB Groningen, The Netherlands

**Keywords:** molecular pathways, oncolytic virus, pattern recognition receptors, transcription factors

## Abstract

Oncolytic virotherapy is a promising immunotherapy approach for cancer treatment that utilizes viruses to preferentially infect and eliminate cancer cells while stimulating the immune response. In this review, we synthesize the current literature on the molecular circuits of immune sensing and response to oncolytic virotherapy, focusing on viral DNA or RNA sensing by infected cells, cytokine and danger-associated-signal sensing by neighboring cells, and the subsequent downstream activation of immune pathways. These sequential sense-and-response mechanisms involve the triggering of molecular sensors by viruses or infected cells to activate transcription factors and related genes for a breadth of immune responses. We describe how the molecular signals induced in the tumor upon virotherapy can trigger diverse immune signaling pathways, activating both antigen-presenting-cell-based innate and T cell-based adaptive immune responses. Insights into these complex mechanisms provide valuable knowledge for enhancing oncolytic virotherapy strategies.

## 1. Introduction

Oncolytic virotherapy is a novel class of immunotherapy for cancer treatment. It employs the use of natively occurring or genetically modified viruses with the ability to preferentially infect and eliminate cancer cells [1]. Virus-induced infection and cancer cell death further initiate a cascade of events that not only reduce the tumor burden but also stimulate potent immune responses. This dual mode of action is regulated through the release of infection-related danger signals, activating diverse innate and adaptive immune pathways [2]. The intricate network of molecular signals and pathways, comparable to a ‘molecular circuit’, coordinates the activation of the immune response against cancer cells.

At the molecular level, infection by oncolytic viruses triggers the signaling pathways of the immune response through the release of diverse immunogenic signals. For instance, viruses can trigger target cells directly through pathogen-associated molecular patterns (PAMPs) or indirectly through infection-induced cell death, releasing cytokines or danger-associated molecular patterns (DAMPs) [3,4]. These signals are sensed at a molecular level by the infected cell or neighboring immune cells. Upon detecting the presence of PAMPs or DAMPs, these responder cells release supplementary cytokines and chemokines, fostering a microenvironment conducive to robust antitumor immune responses [5]. As oncolytic virotherapies differ in genetic design and type, specific virus–cell interactions in terms of molecular sensing and response may shape immune-associated signaling in the tumor. Therefore, a better understanding of specific molecular triggers induced by specific viruses can potentially help in optimizing their efficacy. As an illustration, talimogene laherparepvec (T-VEC) is an FDA- and EMA-approved oncolytic virotherapy for advanced melanoma [6,7]. T-VEC employs a genetically modified herpes simplex virus encoding granulocyte-macrophage colony-stimulating factor (GM-CSF). GM-CSF, a key cytokine, plays a crucial role in dendritic cell activation and antigen presentation for effective anticancer T cell responses. Thus, understanding the molecular pathway of GM-CSF-induced immune responses has been crucial for optimizing T-VEC’s design and efficacy. Following this rationale, various virotherapy candidates are currently being engineered to induce an optimal immune response in preclinical and clinical stages [8,9,10].

In the present review, we aim to provide a comprehensive synthesis of the current literature surrounding the molecular circuits of immune sensing and response to oncolytic virotherapy. To focus on molecular signaling, our review will not assess additional factors that may influence the efficacy of virotherapy, such as virus tropism, dosage/administration methods, or patient-specific factors like genetic variations [8,9,11]. Moreover, we will not focus on virus-induced signaling that leads to therapeutic resistance, as this has been reviewed earlier [12,13]. By exploring the diverse pathways triggered by oncolytic viruses and understanding the end response factors, our review aims to summarize key mechanisms associated with the induction of immune responses. To achieve this, we will review studies providing evidence on how oncolytic viruses and related signals are sensed by (i) infected cells, (ii) neighboring immune cells, and how this subsequently leads to (iii) the activation of the immune response in the tumor. Additionally, we review how various oncolytic viruses, distinguished by their genetic composition, uniquely activate the immune system.

## 2. Molecular Sensing and Response to Oncolytic Viruses by Infected Cells

Cells, cancerous or healthy, utilize intricate molecular mechanisms to detect and respond to infection with oncolytic viruses. The extracellular sensing of virus particles or intracellular sensing of viral genetic material, such as double-stranded (dsRNA) or single-stranded (ssRNA) RNA or cytoplasmic DNA molecules, has been widely studied and found to induce innate immune responses in the tumor [3,4,10]. Cellular pattern recognition receptors, including toll-like receptors (TLRs), retinoic acid-inducible gene I (RIG-I)-like receptors (RLRs), and cyclic GMP-AMP synthase (cGAS), play a primary role in the process of sensing and signaling in the response to oncolytic viruses [3,4].

### 2.1. Extracellular Sensing of Oncolytic Viruses

The extracellular sensing of oncolytic viruses by toll-like receptors (TLRs) present on the cell surface, such as TLR2 or TLR4, plays a pivotal role in stimulating immune responses. For instance, TLR4-mediated sensing of vesicular stomatitis virus (VSV) by cancer cells leads to the activation of MyD88 signaling, which results in the induction of type-1 interferon signaling, tumor and lymph node infiltration of T cells and dendritic cells, and overall anticancer immunity [14]. Similarly, the activation of TLR4 by oncolytic adenoviruses has also been shown to promote systemic and specific antitumor immunity upon therapy [15]. Additionally, oncolytic adenoviruses have been demonstrated to activate TLR2 on cancer cells, leading to MyD88-dependent interferon responses and subsequent immune activation [16]. Direct TLR2-mediated activation of NK cells by oncolytic herpes simplex virus has also been shown to promote their anticancer function [17,18]. A combined activation of TLR4 and TLR2 by cowpea mosaic virus has been shown to significantly promote local and systemic antitumor immunity [19]. Although TLR-mediated sensing of oncolytic viruses is generally favorable, it may also induce antiviral immunity and undermine efficacy. For example, vaccinia virus naturally activates TLR2 to induce antiviral antibody responses. In such cases, the blocking of TLR2 activation reduces antiviral antibodies, enhancing virus infection and therapeutic efficacy [20]. Taken together, these findings underscore the critical role of TLR-mediated sensing of extracellular viruses to induce immune responses in the context of cancer therapy.

### 2.2. Sensing Viral RNA

Sensing and responding to RNA-based oncolytic viruses is a multifaceted process crucial for the initiation of immune responses within host cells [3]. The cytoplasmic sensors RIG-I and MDA5 play pivotal roles in detecting both single-stranded and double-stranded RNA within infected cells (Figure 1A). The activation of these sensors elicits a cascade of events leading to the activation of transcription factors like nuclear factor kappa-light-chain-enhancer of activated B cells (NFΚB1) and interferon regulatory factors 3 and 7 (IRF3 and IRF7) [21,22]. Consequently, the expression of type-1 interferons-α and β (IFNα, IFNβ), pro-inflammatory cytokines like interleukin-18 and -12 (IL-18 and IL-12), tumor necrosis factor (TNF), and chemokines like C-X-C motif chemokine ligand 10 (CXCL10) is induced, contributing to the establishment of an antiviral state within the host cell. For instance, upon infection with oncolytic alphaviruses, proteins like RIG-I and TNF receptor-associated factor-6 (TRAF-6) are engaged, leading to the amplification of antiviral responses. Consequently, pro-inflammatory cytokines, including IL-1beta, TNF-alpha, IL-6, and CXCL9, are expressed, enhancing the antitumor milieu and suppressing interferon-stimulated genes (ISGs) partly through zinc-finger antiviral protein (ZAP) expression [23,24,25,26,27,28,29,30]. Similarly, infection with coxsackievirus activates RIG-I, triggering both innate and adaptive immune responses against tumors. This activation leads to the upregulation of IFN-inducible genes and Th1-associated chemokines, facilitating effector T cell recruitment to the tumor microenvironment [31]. In hormone-refractory prostate cancer, Sendai virus also triggers the activation of RIG-I, leading to the upregulation of IFN-related genes and subsequent activation of the Janus kinase and signal transducer and activator of transcription proteins (JAK/STAT) pathway, ultimately inducing apoptosis in cancer cells [32]. Oncolytic reovirus is also known to trigger RIG-I and MDA5 to induce interferon signaling [33]. Virus-induced oncolysis mediated by the RIG-I signaling pathway, can also occur by upregulation of TNF-related apoptosis-inducing ligand, making it a promising target for cancer therapy [34]. This indicates the importance of viral RNA sensing pathways as a key mechanism to induce immune responses in the tumor.

### 2.3. Sensing Viral DNA

Various cellular sensors are also involved in sensing the DNA of oncolytic viruses. Cytoplasmic DNA sensing is facilitated by cGAS and Z-DNA binding protein 1 (ZBP1). Upon the detection of cytoplasmic DNA, cGAS catalyzes the production of cyclic GMP-AMP (cGAMP), initiating downstream signaling events that culminate in the activation of transcription factors NFΚB1, IRF3, and IRF7 (Figure 1B) [37,38]. This results in the expression of type-1 IFNs (IFNα, IFNβ), pro-inflammatory cytokines (IL6), and chemokine motif ligands 5 and 4L1 (CCL5, CCL4L1), bolstering the cellular defense against viral invasion. Similarly, ZBP1 activation leads to the induction of IFNs (IFNα, IFNβ), contributing to the antiviral response elicited upon cytoplasmic DNA detection (Figure 1C). The cytoplasmic sensing of viral signals has been described to potentially induce innate immune responses. For example, cGAS-mediated sensing and activation of immune responses are crucial for the therapeutic efficacy of oncolytic herpes virus [35]. Additionally, it has been shown that the epigenetic downregulation of cGAS-mediated signaling in ovarian cancers also leads to a downregulation of cytokine expression and related immune responses [16]. In this case, compensatory signaling mechanisms such as those activated by dsRNA through RIG-I/MDA5, remain largely unaffected and generate an immune response [16].

The recognition of methylated (CpG) DNA by endosomal toll-like receptor 9 (TLR9) activates the transcription factor NFκB1 (Figure 1D), triggering the expression of pro-inflammatory cytokines (TNF, IL1B, IL6, IL12A) and chemokines (CXCL8, CCL5, CCL3L3, CCL4L1) [21,39]. TLR-9-mediated recognition of viral signals can potentially induce immune responses in the tumor. This has inspired the development of genetically engineered oncolytic parvovirus that induces TLR-9 activation [36]. Here, the incorporation of immunostimulatory CpG motifs into parvoviruses variants like JabCG1 and JabCG2 has been found to boost their adjuvant capacity [36]. These variants trigger TLR-9-mediated signaling, leading to enhanced immunogenicity in animal models of cancer. Notably, JabCG2 demonstrated superior antitumor activity, inducing markers of cellular immunity and dendritic cell activation, thus reducing metastatic rates compared to other treatments. Alternatively, the loss of TLR9-mediated virus recognition has been shown to cause dysfunctional innate immune responses against oncolytic adenoviruses [16].

## 3. Molecular Sensing and Response to Infected Cells

Oncolytic virus-infected cells become active contributors to the intricate molecular signaling within the tumor microenvironment. A myriad of immunostimulatory molecules are released by virus-infected or dying tumor cells due to the diverse signaling pathways triggered by oncolytic viruses [40,41,42,43,44,45,46,47].

### 3.1. Sensing Cytokines from Infected Cells

The production of cytokines, particularly interferons, by infected cells serves as a critical component of the antiviral defense mechanism. One of the pivotal mechanisms involves the recognition of extracellular interferon by interferon-alpha receptors (IFNAR) on adjacent cells (Figure 2A) [48]. This interaction initiates a cascade of events that includes the activation of transcription factors STAT1, STAT2, or IRF9, ultimately leading to the induction of various genes such as CXCL9, CXCL10, CXCL11, and major histocompatibility complex-I and II (MHC-I, and MHC-II) [49]. Therefore, interferons not only exert direct antiviral effects but also facilitate the activation of neighboring immune and other stromal cells. For instance, oncolytic adenoviruses and alphaviruses induce interferon responses through various mechanisms. Adenoviruses trigger interferon signaling by activating the STING pathway and promoting the release of type-1 IFNs (IFNα, IFNβ) in infected tumor cells, which subsequently stimulate an antitumor immune response [50]. Similarly, alphaviruses induce interferon responses by upregulating autophagy and endoplasmic reticulum stress, leading to enhanced therapeutic efficacy through increased apoptosis and reduced tumor proliferation [51,52].

Although important for immune activation, interferon signaling can also undermine the therapeutic efficacy of oncolytic viruses. Multiple reviews on this phenomenon have explained that infection with various oncolytic viruses leads to the induction of a wide array of antiviral proteins in the tumor, which can potentially hinder viral replication and tumor cell oncolysis [12,13]. Researchers have tried to overcome this challenge by either selecting or engineering oncolytic viruses that maintain their function despite interferon signaling. For example, the replication of oncolytic reovirus has been demonstrated to remain unaffected even in the presence of interferon signaling and favorably promoting robust innate immune responses [33]. The engineering of various interferon-resistant oncolytic viruses has also proved to be an effective strategy to benefit from interferon-mediated innate activation of immune responses while maintaining viral replication and infection [12,53,54]. Alternatively, inhibition of interferon signaling using small-molecule inhibitors like ruxolitinib has also been shown to improve the therapeutic efficacy of various oncolytic viruses [55,56].

### 3.2. Sensing Pathogen-Associated Signals from Infected Cells

Molecules such as viral PAMPs (e.g., nucleic acids) serve as potent danger signals, triggering the activation of innate immune receptors in neighboring cells, thereby amplifying the antiviral response. For example, endocytosed or phagocytosed debris containing viral RNA is detected by Toll-like receptors-3, 7, or 8 (TLR-3, TLR7, or TLR-8) within antigen-presenting cells like plasmacytoid dendritic cells (pDCs) and macrophages [39]. The recognition of double-stranded viral RNA in the endosomes is mediated through TLR3 (Figure 2B). Upon encountering dsRNA, TLR3 activates a signaling cascade leading to the activation of transcription factors IRF3 and IRF7 [57]. Consequently, the expression of pivotal immune mediators such as interferons (IFNα and IFNβ), as well as various chemokines (CXCL9, CXCL10, CXCL11) and co-stimulatory molecules like clusters of differentiation 40, 80, or 86 (CD40, CD80, CD86), is induced, orchestrating a robust antiviral response. Single-stranded RNA derived from oncolytic viruses is sensed by endosomal TLR7 and/or 8 (Figure 2B,C) [21]. The activation of these receptors triggers downstream signaling pathways, culminating in the activation of the transcription factor NFΚB1. This activation prompts the expression of pro-inflammatory cytokines (TNF, IL1B, IL6, IL12A) and chemokines (CXCL8, CCL5, CCL3L3, CCL4L1), fostering an inflammatory milieu conducive to innate immune responses. This orchestrated response enhances the immune surveillance against viral infections and contributes to the shaping of an immune response in the tumor milieu. For example, Newcastle disease virus infection activates various signaling mechanisms, including TLR-signaling in tumors to regulate immune response and tumor susceptibility markers [58]. Oncolytic parvovirus exploits TLR signaling to induce human immune responses, enhancing dendritic cell maturation and stimulating NFκB-dependent activation of the adaptive immune system, thereby priming immune responses against tumors [59]. It has also been shown that immune responses induced by viral infection also synergize to enhance oncolysis. For example, IL-24 enhances apoptosis induced by influenza A virus via the TLR3 and caspase-8 pathways, sensitizing cancer cells to TLR-mediated apoptosis [60]. Similarly, measles-virus-based therapies induce oncolysis by activating plasmacytoid dendritic cells, resulting in the production of interferon-alpha and cross-presentation of tumor antigens, thereby facilitating antigen-specific immune responses against tumors [61]. Alternatively, some variants of parvoviruses escape TLR/RIG-I mediated immune sensing by tumor cells, resulting in lower IFN production and subsequent immune responses [62].

### 3.3. Sensing Danger-Associated Signals from Infected Cells

The process of viral replication and cell lysis leads to the release of cellular DAMPs from dying cancer cells, acting as an endogenous danger signal. For example, extracellular high-mobility group box 1 (HMGB1) is a DAMP sensed by neighboring antigen-presenting cells (Figure 2D) [63]. Interaction with Toll-like receptor 4 (TLR4) sets off a signaling cascade that involves the activation of transcription factors NFΚB1 or IRF5 [64]. This activation, in turn, leads to the expression of a plethora of pro-inflammatory and immunomodulatory molecules including IFNα, IFNβ, TNF, IL1B, IL 6, IL 12A, CXCL8, CCL5, CCL3L3, and CCL4L1. Thus, the recognition of and response to DAMPs further amplify the immune response against viral infections and contribute to the overall activation of immune responses in the tumor microenvironment. For instance, upon infection with oncolytic adenoviruses, tumor cells release DAMPs such as HMGB1 and ATP, which activate dendritic cells and promote tumor-specific T cell responses, contributing to the antitumor immune response [50]. Similarly, oncolytic parvovirus induces cell death in glioma cells, activating dendritic cells and microglia, thereby breaking tumor tolerance and inducing long-term memory responses against tumors [35].

## 4. Molecular Signaling Response for Immune Activation

The tumor microenvironment is composed of cancer cells, immune cells, fibroblasts, blood vessels, and extracellular matrix components. In response to signals from viruses or infected cells, various components of the tumor microenvironment engage in intricate molecular signaling pathways [5], culminating in the activation of both innate and adaptive immune responses [52]. Within this environment, antiviral tumor immune responses can occur, where the immune system targets viruses present within tumor cells to hinder viral replication and potentially support tumor growth [4,12,13]. Alternatively, molecular signaling may also contribute to antitumor effects, which encompass a range of biological mechanisms aimed at inhibiting or regressing the tumor [2,5,10]. Among these mechanisms are immune-mediated destruction of tumor cells, inhibition of tumor cell proliferation, induction of apoptosis, and suppression of tumor angiogenesis [65,66]. Central to these processes are tumor-specific T cell responses, where T cells recognize and target tumor antigens displayed on the surface of tumor cells, leading to their destruction and potentially aiding in the control or elimination of the tumor [66,67]. Together, these interactions influence the outcomes of oncolytic virotherapy by regulating the interplay between the immune system and tumor cells in the regulation of tumor growth and progression.

### 4.1. Innate Immune Signaling

The activation of innate immune responses by oncolytic viruses plays a pivotal role in eliciting antitumor effects [4]. These include the recruitment and activation of innate immune cells, including neutrophils, macrophages, natural killer cells, and dendritic cells, which aid in antigen presentation and maintaining a pro-inflammatory environment [4]. Furthermore, this may also lead to the recruitment and activation of T cells with cytotoxic antitumor activity [2].

Innate immune signaling is regulated through diverse chemokines recognized by their respective chemokine receptors, triggering the activation of transcription factors such as Forkhead box O (FOXO), NFΚB1, and STAT, among others (Figure 3A). This activation results in the expression of a wide array of genes promoting immune cell proliferation, differentiation, and migration to sites of action [4,5]. For example, IL-2 or IL-4, sensed by their receptors, IL-2 receptor and IL-4 receptor, respectively, activate the transcription factor STAT5, leading to the autocrine expression of IL-2, IL-4, IL-2 receptor, IL-4 receptor, and other cytokines (Figure 3B) [68,69]. Similarly, interferon-gamma (IFNγ), sensed by interferon-gamma receptor (IFNGR), triggers the activation of transcription factors STAT1 and STAT4, inducing the expression of IFNγ, T-box transcription factor (T-bet), and interleukin-12 receptor (IL12RB1) (Figure 3C) [70]. Taken together, this intricate network of signaling pathways orchestrates a pro-inflammatory immune response that promotes the recruitment and proliferation of immune cells in the tumor, facilitating antigen presentation and immune-mediated killing of cancer cells. For example, replication-competent Sendai viruses (rSeV) have been observed to activate dendritic cells through the RIG-I pathway, inducing the production of type I IFNs, which contribute to their antimetastatic effects [71]. Similarly, measles virus exploits its receptors CD150 and CD46 on tumor cells to trigger the immune response, leading to increased IFNγ levels and a favorable immune milieu for tumor clearance [72]. Vesicular stomatitis virus induces type-I IFN responses, enhances dendritic cell maturation, and promotes antigen presentation, thus facilitating the initiation of adaptive immune responses against tumors [14,73,74].

### 4.2. Adaptive Immune Signaling

In addition to innate immune activation, oncolytic viruses also stimulate adaptive immune responses, which are crucial for sustained antitumor effects. In the context of antigen-specific or target-specific immune activation, viral or tumor antigens presented by MHC-I and II molecules and sensed by T cell receptors initiate signaling cascades [75,76]. This activation leads to the engagement of transcription factors like extracellular signal-regulated kinases 1 and 2 (ERK1, ERK2) and nuclear factor of activated T cells (NFAT), prompting the expression of perforin, granzyme, and FAS-ligand (FAS-L), ultimately resulting in the killing of target cells (Figure 4A). Similarly, antigens presented by MHC molecules and sensed by T cell receptors activate transcription factors NFΚB1, NFATC1, protein c-Fos (FOS), and protein c-Jun (JUN), resulting in the expression of IFNγ, T-bet, TNF, GM-CSF, IL-2, IL-4, IL-5, IL-10, and IL-13 (Figure 4B) [75,76]. Moreover, the loss or downregulation of MHC molecules or the presence of unconventional MHC molecules, recognized by NK cell receptors, activate transcription factors ERK1, ERK2, and NFAT, leading to the expression of perforin, granzyme, and FAS-ligand, thereby facilitating target cell killing (Figure 4C) [77,78,79]. Overall, it has been observed that various oncolytic virotherapies facilitate the activation of antigen-specific immune responses. For instance, Newcastle disease virus (NDV) oncolysis induces immunogenic cell death, leading to tumor infiltration by effector T lymphocytes and long-term tumor-specific immunological memory responses [80,81]. H-1 parvovirus selectively activates helper and not regulatory CD4+ T cells, thus demonstrating its potential as an anticancer treatment without exhibiting immunosuppressive effects [82]. The activation of helper CD4+ T cell responses by H-1 parvovirus further underscores its immunotherapeutic potential. These findings highlight the dual role of oncolytic viruses in triggering both innate and adaptive immune responses, thereby enhancing their efficacy as cancer therapeutics.

## 5. Strategies to Exploit Immune Signaling in Favor of Oncolytic Virotherapy

Various avenues have been explored to improve the safety and efficacy of oncolytic virotherapy. Inspired by the molecular mechanisms employed by virotherapy to trigger immune response pathways, efforts have been made to boost their immunogenic potential. These strategies include engineering viruses to enhance immune signaling, targeting tumors specifically, and employing combinatorial approaches with immunomodulatory agents or other viral therapies to amplify antitumor effects.

### 5.1. Engineering Viruses to Trigger Immune Signaling

Various oncolytic viruses have been engineered to bolster the immune response against cancer [83,84]. One approach involves encoding cytokines and chemokines within these viruses to enhance immune stimulation. For instance, adenoviruses can be armed with molecules such as GM-CSF and B7-1 to activate dendritic cells and facilitate T cell infiltration into tumors, thereby priming tumor-specific cytotoxic T lymphocytes [85]. Additionally, interleukin-12 expressed by adenoviruses promotes Th1-type immune responses and enhances NK cell and cytotoxic T cell activity [86]. Furthermore, molecules like CCL5 and beta-defensin-2 recruit and activate immune cells within the tumor microenvironment [15,87]. Incorporating CD40 ligand (CD40L) and IL-24 further amplifies immune-mediated tumor cell killing [88,89]. Adenoviruses encoding combinations of cytokines such as IL-12 and IL-18 could synergistically enhance antitumor immunity [90]. In HSV vectors, encoding immunomodulatory cytokines like IL-2, IL-12, and GM-CSF enhances tumor regression mediated by CD4+ and CD8+ lymphocytes [91,92,93]. Methods such as the HSVQuik system expedite the generation of oncolytic HSV vectors expressing immunomodulators for cancer gene therapy [94]. Similarly, oncolytic influenza A viruses armed with immune-stimulating molecules like OX40L exhibit potent oncolytic effects, selectively destroying tumor tissues and enhancing Th1-dominant immune responses [95]. Measles virus strains have demonstrated oncolytic capabilities in hepatocellular carcinoma and glioblastoma cells, inhibiting tumor growth and improving survival rates [96,97,98]. Recombinant Newcastle disease virus strains expressing cytokines such as IL-2, IL-15, or TRAIL stimulate tumor-specific CTL responses, inducing CD4+ and CD8+ T cell proliferation, leading to tumor regression [99,100]. Genetically engineered Sendai virus carrying the IL-2 gene stimulates antitumor effects by modulating immune cell populations [101]. Oncolytic vaccinia viruses, armed with immunomodulatory molecules play a crucial role in activating molecular immune responses against cancer [20,102,103,104,105,106,107,108,109,110]. These viruses enhance antitumor efficacy by stimulating the activation of CD4+ and CD8+ T cells, promoting immune cell infiltration, and augmenting immune-based antitumoral activity through the delivery of chemokines and immunomodulatory antibodies. VSV engineered to encode therapeutic genes like IFN-beta enhances antitumor effects [111], although certain molecules like CD40L may not significantly improve efficacy [112].

### 5.2. Engineering Viruses for Tumor Targeting

Tumor-targeting strategies aim to enhance the immunotherapeutic potential of oncolytic viruses by selectively targeting components of the tumor microenvironment. Utilizing tumor-specific promoters such as CXCR4-promoter allows for tumor-specific transgene expression, driving high expression of therapeutic genes like GM-CSF and B7-1, thereby enhancing immune cell activation and infiltration into tumors [85,113]. Various delivery strategies, including gelatin gel-mediated co-delivery, myeloid-cell mediated delivery, and mesenchymal stem cell-mediated delivery, improve the sustained release and tumor targeting of oncolytic viruses, thereby enhancing their therapeutic efficacy [114,115,116].

### 5.3. Combinatorial Approaches to Boost Viral Immunogenicity

Combination strategies play a crucial role in enhancing the efficacy of oncolytic viruses in cancer therapy. One approach, exemplified by the combination of oncolytic adenoviral therapy with immune checkpoint inhibitors, involves leveraging immune signaling pathways. Adenoviruses engineered to express TNF-alpha and IL-2, when coupled with anti-PD-1 therapy, exhibit augmented tumor control through increased CD8+ T cell infiltration and reduced immunosuppressive cell populations [86,117]. Additionally, combining oncolytic virotherapy with conventional treatments such as gemcitabine shows promise in enhancing oncolytic activity and eliciting systemic antitumor immunity by reducing myeloid-derived suppressor cells and promoting tumor regression [118]. Furthermore, combining oncolytic vaccinia viruses with immune checkpoint blockade amplifies antitumor immunity by increasing effector T cell infiltration, inducing PD-L1 expression, and reducing exhaustion markers [119,120].

Immunomodulatory agents further enhance the efficacy of oncolytic viruses. For instance, the combination of oncolytic virotherapy with the DTA-1 monoclonal antibody enhances tumor growth inhibition by fostering CD8+ T cell accumulation and diminishing regulatory T cells [121]. Similarly, blocking TNFα, which can impede oncolytic herpes virus replication by macrophages and microglia, augments virus replication and improves survival rates in glioblastoma models [122]. Augmenting dendritic cell populations through FMS-like tyrosine kinase ligand (Flt3L) administration improves tumor antigen cross-presentation and CD8+T cell responses, thereby enhancing the efficacy of oncolytic Newcastle disease virus [123]. Finally, synergistic antitumor effects are observed when combining oncolytic virotherapy with other viral therapies. Combining Newcastle disease virotherapy with influenza HA2 gene therapy or viral sensitizer-mediated therapy enhances immune responses and yields heightened antitumor effects [124,125].

## 6. Conclusions

In summary, our review delves into the intricate interplay between viral mechanisms of molecular sensing and the immune response across diverse oncolytic viruses. We systematically reviewed the literature concerning various molecular sensors, transcription factors, and immune signals associated with different oncolytic virotherapies (Figure 5). We summarize a sequential sense-and-response mechanism wherein diverse molecular sensors such as TLRs, RIG-I, and others detect viral genetic material or signals emanating from infected cells, ultimately triggering a cascade of innate and antigen-specific immune responses. This intricate process is finely regulated by the activation of various transcription factors, which in turn induce the expression of immune response genes. Notably, our analysis provides comprehensive insights into these molecular interactions, shedding light on the complex mechanisms underlying the efficacy of oncolytic virotherapies.

## Figures and Tables

**Figure 1 ijms-25-04691-f001:**
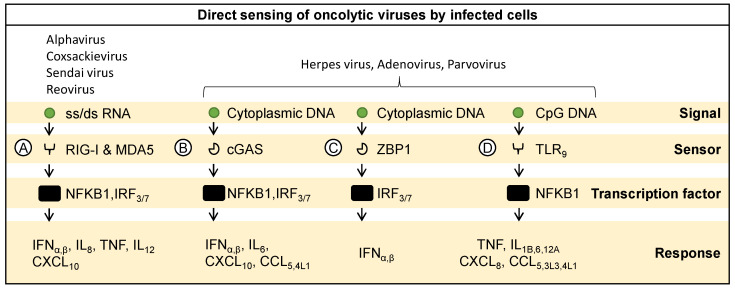
Direct sensing of oncolytic viruses and related immune responses. The figure illustrates essential pathways involved in molecular sensing and response during viral infection. (**A**) Cytoplasmic receptors RIG-I and MDA5 detect single- or double-stranded RNA, activating transcription factors NFΚB1, IRF3, and IRF7, leading to the production of interferons and pro-inflammatory cytokines [23,24,25,26,27,28,29,30,31,32,33]. (**B**) Cytoplasmic DNA sensed by cGAS triggers the activation of transcription factors NFΚB1, IRF3, and IRF7, leading to the expression of key cytokines and chemokines [35]. (**C**) Cytoplasmic DNA detection by ZBP1 results in the activation of transcription factors IRF3 and IRF7, inducing the expression of interferons. (**D**) Methylated CpG DNA recognition by endosomal TLR9 activates NFΚB1, leading to the expression of pro-inflammatory mediators [16,36].

**Figure 2 ijms-25-04691-f002:**
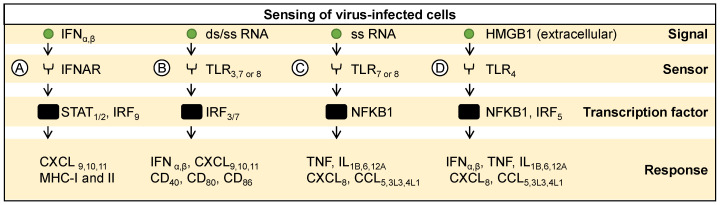
Sensing of immunostimulatory signals released by infected cells. (**A**) Interferon alpha and beta are recognized by IFNAR receptors, inducing expression of various antiviral genes and antigen presentation. Endocytosed or phagocytosed debris containing viral PAMPs (e.g., double- or single stranded-RNA) activates transcription factors like (**B**) IRF3 and IRF7 or (**C**) NFΚB1 in antigen-presenting cells, leading to pro-inflammatory cytokine production and upregulated antigen presentation. (**D**) HMGB1, a DAMP, is sensed by TLR4, triggering immune responses.

**Figure 3 ijms-25-04691-f003:**
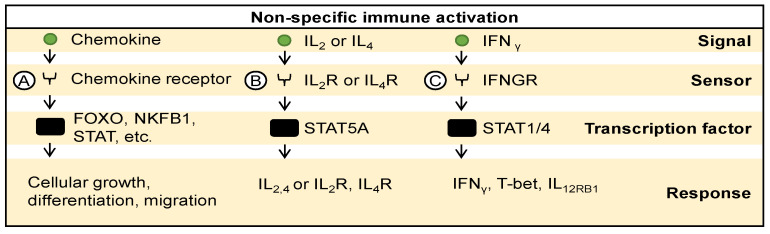
Non-specific immune activation. (**A**) Chemokines, recognized by their respective receptors, activate various transcription factors promoting immune cell proliferation, differentiation, and migration. For example, (**B**) IL2 and IL4 activate STAT5, inducing autocrine and paracrine expression of cytokines and receptors. (**C**) Similarly, IFN gamma activates STAT1 and STAT4, inducing expression of immune-related genes. Overall, these pathways regulate immune responses against infections and cancer.

**Figure 4 ijms-25-04691-f004:**
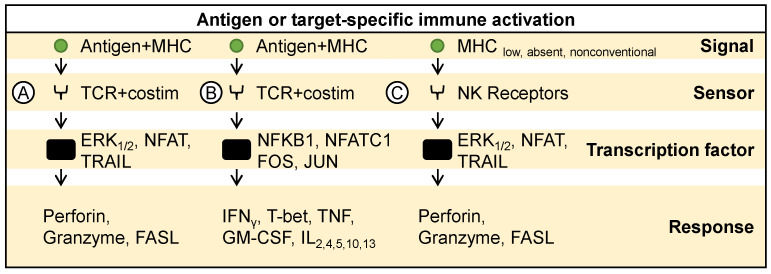
Target-specific immune activation. Antigenic exposure to T cells activates various transcription factors, promoting (**A**) cytotoxic activity towards target cells and (**B**) release of various pro-inflammatory molecules promoting antigen presentation and cancer killing. (**C**) Similarly, NK cells recognize loss or downregulation of MHC molecules on target cells and activate cytotoxic activity towards target cells.

**Figure 5 ijms-25-04691-f005:**
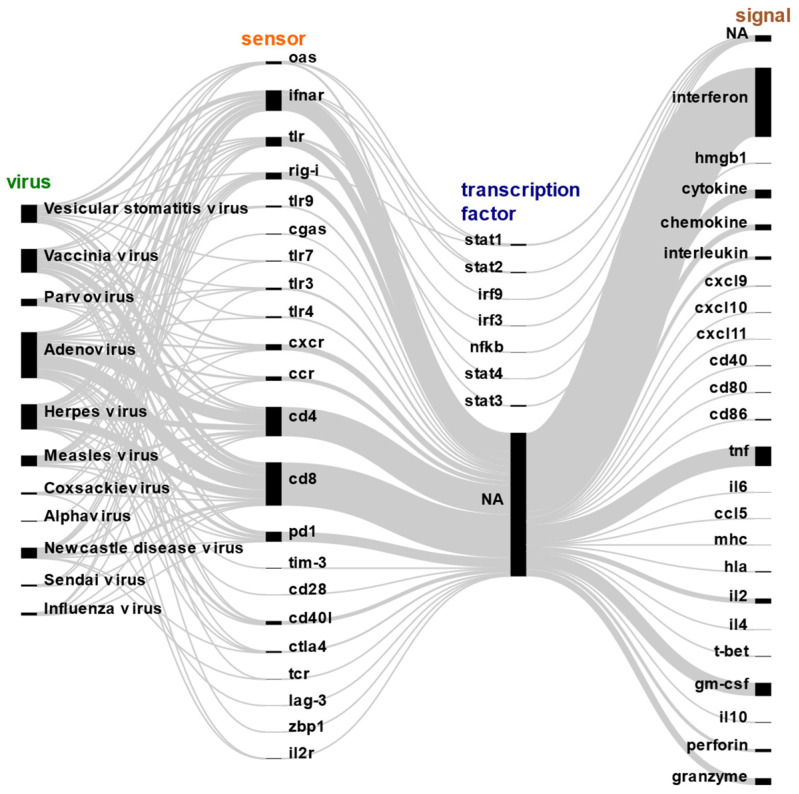
Literature-based association of viral mechanisms of molecular sensing and immune response to diverse oncolytic viruses. We collected abstracts of scientific articles from PubMed and looked for the presence of molecular evidence on various sensors, transcription factors, and immune signals that were associated with different oncolytic virotherapies. In a sequential sense-and-response manner, diverse molecular sensors, like TLRs, RIG-I, etc., detect virus genetic material or signals from infected cells to ultimately induce various innate and antigen-specific immune responses. This is regulated by the activation of various transcription factors that induce the expression of immune response genes. NA = not available. Each line in the figure represents connections between different components involved in the molecular sensing and immune response to oncolytic viruses. These connections visually illustrate how various sensors, transcription factors, and immune signals influence each other in initiating and regulating the immune response. For instance, lines can associate the activation of transcription factors by sensors or the regulation of immune response genes by transcription factors.

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
