# Peer review of "Molecular Circuits of Immune Sensing and Response to Oncolytic Virotherapy"

_ijms, 2024, doi:10.3390/ijms25094691_

Round 1

Reviewer 1 Report

Comments and Suggestions for Authors

In this review, authors Bhatt and Daemen aim to provide a comprehensive synthesis of the current literature surrounding the molecular circuits of immune sensing and response to oncolytic virotherapy. The authors share information on molecular sensing of viral RNA and DNA (section 2), response to released cytokines by infected cells (section 3), activation of immune signaling pathways (section 4), and strategies to exploit this for oncolytic virotherapy (section 5). Much of this work (sections 2-4) really is a survey of these topics as it relates to all viruses, not specifically to oncolytic viruses necessarily, and it isn’t until section 5 that oncolytic viruses, or the specific tumor environment, are consistently prioritized. There is no doubt that a lot of work went into drafting this review, and processing all of the articles contained within. However, if a reader is interested in understanding innate signaling or molecular sensing of viruses, other reviews provide a clearer, more comprehensive analysis of the field. The strength of this article lies in the content found in section 5, which describe strategies to exploit immune signaling in favor of oncolytic virotherapy. I offer the following major and minor comments for consideration.

Major comments:

1. Concerns with citations: There are repeated instances in this manuscript that could benefit from a citation. For instance, lines: 54-55, lines 65-81 (numerous examples and zero citations!), lines 92-95, lines 141-146, etc. A careful review of the text to incorporate citations appropriately, as well as a careful review of citation format in the references section should be conducted. Additionally, 57/113 references presented in this paper were at least 10 years old. In a field that moves quickly, the authors might consider trying to incorporate newer, more relevant work.

2. Organization and flow: Cellular signaling pathways, cytokine responses, and molecular sensing pathways are complex and often described using acronyms. The authors should introduce each section with a general overview of how a system functions. This should be followed by examples (not just listed, but described and providing context for). By doing so, the authors could increase the readability and usefulness of this review. For instance, in Section 2, lines 65-81, the authors introduce how cells detect and respond to infection with oncolytic viruses (although what is described is true for viruses in general). However, instead of a comprehensive overview exposing areas at each stage of the life cycle, the authors list the genome types, providing example viruses for each, but only go on to explore DNA viruses in detail. Importantly, the authors only discuss sensing of genomes inside of cells, completely ignoring detection of extracellular virus. Overall, this paragraph and others are cumbersome and a more direct, comprehensive overview could be provided. 

3. Figures: The figures in this manuscript are very basic. I would encourage the authors to minimally add a row to highlight examples of oncolytic viruses that are sensed/interfere/activate each pathway. Incorporated into this should be citations immediately following each virus so that readers can easily identify relevant work. Additionally, many papers similar to this have a separate table of all oncolytic viruses breaking down how each subverts/benefits from different sensing pathways, etc.

4. Section 3: Sensing and implications for efficacy: In this section, the cell’s sensing of oncolytic viruses (read all viruses) contributes to molecular signaling within the tumor microenvironment. The authors discuss the importance of producing interferon as a critical component of the antiviral defense mechanism. However, no mention is made of how this will impact oncolytic therapy. For instance, if oncolytic viruses activate interferon, which is generally regarded as detrimental to the virus, the authors should speak to the potential limitations in spread and infectivity of progeny virions during oncolytic therapy. Does interferon limit its effectiveness? Importantly, there are some oncolytic viruses, including reoviruses, that actually benefit from this response. However, the authors make no mention of these viruses in this review.

5. Explanation of terms and interpreting findings. If this were a primary research article, not defining the following terms would be expected. However, given that this is a review article with the obligation to potentially inform readers new to the field, terms such as tumor microenvironment, antiviral tumor immune responses, antitumor effects, tumor-specific T cell responses, should be defined/described at a minimum, preferably with elaboration. For instance, lines 232-233: “Activation of innate immune responses by oncolytic viruses plays a pivotal role in eliciting antitumor effects.” This sentence would be far more useful to the reader if the authors described what those antitumor effects were. Or, Lines 237-240: “For example, IL-2 or IL-4 sensed by their receptors, IL-2 receptor and IL-4 receptor respectively, activate the transcription factor STAT5, leading to the autocrine expression of IL-2, IL-4 , IL-2 receptor, IL-4 receptor, and other cytokines”. This sentence would be more useful if it instead ended with “resulting in the following changes: activation of T cells, etc….” One objective of a review article should be to interpret findings and place them in the context of the field. 

Minor Comments:

1. Lines 41-50: The example described in the introduction is from 2014. A newer example would be more relevant.

2. Lines 73-75: The authors describe that viruses with DNA genomes may exploit viral protein and machinery to transport DNA from the nucleus to the cytoplasm, facilitating virus replication and detection by cytoplasmic DNA sensors. Why would a DNA Virus want to be detected by cell DNA sensors? Generally, viruses including oncolytic viruses, try to avoid detection. Addressing this would be useful.

3. Section 3.2. It is unclear if the authors are discussing a response in infected cells, in cells neighboring cells that are part of the tissue or APCs. In general, I do not think immune cells and APCs are considered neighboring cells, rather neighboring cells would be those adjacent to infected cells in the tissue. Since section 3.2 deals primarily with PAMPs, the authors might consider moving this content to section 2 (molecular sensing).

4. Section 5.2 (Engineering viruses for tumor targeting) should explore mechanisms that promote virus trafficking to tumor environments. Lines 332-334 describe how oncolytic adenovirus repolarizes the microenvironment to a pro-inflammatory state and, although interesting, doesn’t align with the objective of the section.

Reviewer 2 Report

Comments and Suggestions for Authors

The manuscript from Bhatt et al presents a comprehensive review of the molecular mechanisms involved in immune sensing and response to oncolytic virotherapy. The authors have done a commendable job synthesizing a broad array of literature to illustrate how different oncolytic viruses interact with the host's immune system. Overall, the manuscript is well-structured, especially in delineating the complex interactions between oncolytic viruses and the immune system, but two aspects that could be improved:

1.     The authors listed most oncolytic virotherapy strategies and the underlying mechanisms, but without clarifying the limitations of each therapeutic strategy and the potential methods to overcome them based on the knowledge of immune sensors.

2.     All the figures in the manuscript are well-integrated and effective, however, it is not necessary to use subscripts, such as CXCL10. If the authors want to save space by using subscripts, please keep the form consistent, in figure 1, the formation is: CCL5,3L3,4L1, but in figure 2, it changed to CD40, CD80,CD86, but not CD40,80,86.

Round 2

Reviewer 1 Report

Comments and Suggestions for Authors

The authors have addressed all of my concerns with the initial submission. No further comments.